# What Does the Future Hold? Health-Related Quality of Life 3–12 Years Following a Youth Sport-Related Knee Injury

**DOI:** 10.3390/ijerph18136877

**Published:** 2021-06-26

**Authors:** Christina Y. Le, Clodagh M. Toomey, Carolyn A. Emery, Jackie L. Whittaker

**Affiliations:** 1Faculty of Rehabilitation Medicine, University of Alberta, 8205 114 Street, 2-50 Corbett Hall, Edmonton, AB T6G 2G4, Canada; cyle@ualberta.ca; 2Arthritis Research Canada, Milan Ilich Arthritis Research Centre, 5591 No. 3 Road, Richmond, BC V6X 2C7, Canada; 3School of Allied Health, Faculty of Education & Health Sciences, University of Limerick, V94 T9PX Limerick, Ireland; clodagh.toomey@ul.ie; 4Health Research Institute, University of Limerick, V94 T9PX Limerick, Ireland; 5Sport Injury Prevention Research Centre, Faculty of Kinesiology, University of Calgary, 2500 University Drive NW, Calgary, AB T2N 1N4, Canada; caemery@ucalgary.ca; 6Departments of Pediatrics and Community Health Sciences, Cumming School of Medicine, Health Sciences Centre Foothills Campus, University of Calgary, 3330 Hospital Drive NW, Calgary, AB T2N 4N1, Canada; 7Alberta Children’s Hospital Research Institute, University of Calgary, Room 293, Heritage Medical Research Building, 3330 Hospital Drive NW, Calgary, AB T2N 4N1, Canada; 8McCaig Institute for Bone and Joint Health, Cumming School of Medicine, University of Calgary, HRIC 3A08, 3280 Hospital Drive NW, Calgary, AB T2N 4Z6, Canada; 9Department of Physical Therapy, Faculty of Medicine, University of British Columbia, #223 212 Friedman Building, 2177 Wesbrook Road, Vancouver, BC V6T 1Z3, Canada

**Keywords:** osteoarthritis, pain, physical activity, prevention

## Abstract

Knee trauma can lead to poor health-related quality of life (HRQoL) and osteoarthritis. We aimed to assess HRQoL 3–12 years following youth sport-related knee injury considering HRQoL and osteoarthritis determinants. Generic (EQ-5D-5L index, EQ-VAS) and condition-specific (Knee injury and Osteoarthritis Outcome Score quality of life subscale, KOOS QOL) HRQoL were assessed in 124 individuals 3–12 years following youth sport-related knee injury and 129 uninjured controls of similar age, sex, and sport. Linear regression examined differences in HRQoL outcomes by injury group. Multivariable linear regression explored the influence of sex, time-since-injury, injury type, body mass index, knee muscle strength, Intermittent and Constant Osteoarthritis Pain (ICOAP) score, and Godin Leisure-Time Exercise Questionnaire (GLTEQ) moderate-to-strenuous physical activity. Participant median (range) age was 23 years (14–29) and 55% were female. Injury history was associated with poorer KOOS QOL (−8.41; 95%CI −10.76, −6.06) but not EQ-5D-5L (−0.0074; −0.0238, 0.0089) or EQ-VAS (−3.82; −8.77, 1.14). Injury history (−5.14; −6.90, −3.38), worse ICOAP score (−0.40; −0.45, −0.36), and anterior cruciate ligament tear (−1.41; −2.77, −0.06) contributed to poorer KOOS QOL. Worse ICOAP score contributed to poorer EQ-5D-5L (−0.0024; −0.0034, −0.0015) and higher GLTEQ moderate-to-strenuous physical activity to better EQ-VAS (0.10; 0.03, 0.17). Knee trauma is associated with poorer condition-specific but not generic HRQoL 3–12 years post-injury.

## 1. Introduction

Traditionally, osteoarthritis has been considered a disease, largely defined by structural findings seen on imaging. However, there is an inconsistent relationship between imaging findings and outcomes associated with the personal (e.g., disability [1], obesity [2], reduced health-related quality of life, HRQoL [3]) and societal burden (e.g., rising healthcare costs, workforce productivity loss [4,5]) of osteoarthritis. We must distinguish between osteoarthritis disease (i.e., pathophysiology) and illness (i.e., a person’s lived experience of a condition) because the features of osteoarthritis illness, rather than osteoarthritis disease, are what drives people to seek healthcare.

Knee joint trauma is highly prevalent [6] and an established risk factor for osteoarthritis disease [7,8,9], but it is unclear if it is also associated with features of osteoarthritis illness, including reduced HRQoL. Health-related quality of life is a multifactorial construct that encompasses the physical, psychological, and social domains of health and is influenced by an individual’s perceptions, experiences, expectations, and beliefs [10]. Following a traumatic knee injury, HRQoL may serve as a valuable indicator of osteoarthritis illness because it represents one’s health across multiple domains. To gain a thorough understanding of HRQoL, we must assess both generic and condition-specific HRQoL. Generic instruments are best for capturing overall HRQoL and allow for comparisons across different demographic groups or medical conditions. On the other hand, condition-specific HRQoL instruments offer a more nuanced understanding of a particular patient group or condition. Previous research has revealed that individuals with osteoarthritis (injury history not specified) report worse generic and condition-specific HRQoL compared to healthy individuals [11,12]. Similarly, there is evidence of poor generic and condition-specific HRQoL of individuals who have experienced an anterior cruciate ligament (ACL) tear at various timepoints [13,14,15].

How and to what extent knee joint trauma contributes to changes in HRQoL is unknown. Female sex [16], older age [17], increased pain [18], less physical activity [19], and obesity [20] are associated with reduced generic HRQoL in healthy populations. Similarly, not returning to sport at the same or higher level and higher body mass index (BMI) are linked to poorer generic HRQoL (Assessment of Quality of Life 8D Utility Instrument; *r*^2^ 0.19), whereas not returning to sport, higher BMI, and subsequent surgery are linked to poorer condition-specific HRQoL (Knee injury and Osteoarthritis Outcome Score quality of life subscale, KOOS QOL; *r*^2^ 0.24) in adults experiencing knee difficulties (e.g., pain, symptoms, functional limitations) 5–20 years post-ACL reconstruction [21]. Notably, many of these characteristics are also established risk factors for structural features of osteoarthritis disease, including female sex [7], older age [7,8], obesity [22,23], and knee extensor muscle weakness [24].

Despite preliminary evidence that HRQoL is impacted following an ACL tear, there is a lack of knowledge about generic and condition-specific HRQoL, and potential osteoarthritis illness, of individuals who have experienced a broader range of knee injuries compared to individuals of similar age, sex, and sport exposure. A better understanding of what factors influence HRQoL of youth and young adults with a history of knee joint trauma is needed to inform future interventions. Therefore, the objective of this study was to assess generic (EQ-5D five-level index score, EQ-5D-5L and EQ-visual analogue scale, EQ-VAS) or condition-specific (KOOS QOL) HRQoL in individuals with a 3–12 year history of a youth sport-related knee injury compared to uninjured controls. To further understand what factors may influence the relationship between injury history and HRQoL, general determinants of HRQoL and osteoarthritis disease (i.e., sex, time since injury, type of injury, BMI, knee extensor strength, knee flexor strength, intermittent knee pain, and self-reported physical activity) were also examined.

## 2. Materials and Methods

### 2.1. Study Design

This study was a cross-sectional analysis of data from the first follow-up (3–12 years post-injury) of the Alberta Youth Prevention of Early Osteoarthritis (PrE-OA) historical cohort study.

### 2.2. Ethics

Ethics approval was granted by the Conjoint Health Research Ethics Board at the University of Calgary, Canada (Ethics ID E-25075). Before testing, all participants provided informed consent/assent and completed a Physical Activity Readiness Questionnaire (PAR-Q, 2002).

### 2.3. Participants

The PrE-OA cohort consists of a convenience sample of individuals who sustained a youth (≤18 years old) sport-related knee injury 3–12 years previously and uninjured controls of similar age (within 12 months), sex, and sport at the time of injury. Information about cohort recruitment, injury diagnosis, and inclusion and exclusion criteria has been described previously [22,25,26]. Briefly, injured and uninjured participants were recruited after being identified from previous cohort studies examining risk factors for sport injury, a university-based sport medicine centre database, or through collaborators and participants. Injured participants sustained a youth sport-related knee injury (clinical diagnosis of a ligament, meniscus, or other intra-articular tibiofemoral or patellofemoral injury) that required medical attention (e.g., physician, physiotherapist) and disrupted sport participation 3–12 years previously. Uninjured controls were included if they reported no previous knee injury resulting in time-loss from sport. Individuals were excluded if they were pregnant; reported non-steroidal anti-inflammatory use, cortisone injection, or other musculoskeletal injury that disrupted sport, school, or work participation within three months prior to testing; or had a diagnosis of other arthritides or any medical conditions that prevented study participation (e.g., neurological conditions).

### 2.4. Procedures

These analyses examined data from the first follow-up (3–12 years post-injury) collected during one testing session at the University of Calgary between 2013–2017 [22,25]. Participants completed a battery of questionnaires (study questionnaire, EQ-5D, KOOS, Intermittent and Constant Osteoarthritis Pain Score, ICOAP, and Godin Leisure-Time Exercise Questionnaire score, GLTEQ) then rotated through testing stations that measured their height, weight, and isometric knee strength. A secure, online database was used to store and manage data (REDCap 8.6.5, Vanderbilt University, Nashville, TN, USA).

A study questionnaire gathered participant information (i.e., age, sex), sport information (i.e., pre-injury main sport, sport participation in the last 12 months), and knee injury details as applicable (i.e., type of injury, injury date, subsequent injury or surgery).

### 2.5. Outcomes

#### 2.5.1. Generic Health-Related Quality of Life

The EQ-5D is a self-reported instrument that measures generic HRQoL [23,27,28]. The EQ-5D is a widely used health utility instrument that consists of two components: the EQ-5D-5L index and EQ-VAS scores. The EQ-5D-5L describes one’s health state and is measured in five dimensions: mobility, self-care, usual activities, pain/discomfort, and anxiety/depression. Participants indicated their health state by selecting one of five levels of responses ranging from no problems to extreme problems for each dimension. Using the Canadian value set [26], health states were converted into EQ-5D-5L index scores which range from −0.148 (worst health status) to 0.949 (best health status). The EQ-VAS evaluates health on a 20 cm vertical visual analog scale with anchors of 0 (worst health you can imagine) and 100 (best health you can imagine). The EQ-5D has been shown to have acceptable reliability [27] and validity [27,28] across musculoskeletal conditions. Although the EQ-5D has not been validated for individuals following a knee injury, it has been previously applied to individuals following ACL reconstruction [29]. The minimal important change (MIC) for Canadian EQ-5D-5L index scores is 0.056 [30] but no MIC for the EQ-VAS in a comparable sample has been established. However, it should be noted that our analyses did not measure change in scores over time.

#### 2.5.2. Condition-Specific Health-Related Quality of Life

The KOOS QOL is one of five subscales of the KOOS [31,32] and assesses knee-specific HRQoL. It consists of four items (awareness of knee problem, lifestyle modification, knee confidence, and overall knee difficulty) scored on a 5-point Likert scale. Subscale scores are converted into a score ranging from 0–100 with higher scores indicating better outcomes. The KOOS demonstrates sufficient internal consistency (pooled Cronbach’s alpha 0.79), test-retest reliability (pooled intraclass correlation coefficient 0.88), and measurement error (pooled standard error of measurement 5.9, pooled smallest detectable change 16.3) in ACL injured samples [31,33]. The MIC for the KOOS QOL subscale is 18.3 in individuals who have undergone ACL reconstruction [34].

#### 2.5.3. Body Mass Index

Body mass index (kg/m^2^) was calculated from weight (to the nearest 0.1 kg) and height (to the nearest 0.1 cm, shoes removed) measurements using a scale and stadiometer (Model 402 KL, Pelstar, McCook, IL, USA).

#### 2.5.4. Isometric Knee Strength

Normalized isometric knee extensor and flexor strength of the injured (index) limb were measured using handheld dynamometry (Model 01163, Lafayette Instrument, Lafayette, IN, USA) [25]. Prior to testing, all examiners were given a written description of testing and scoring. Each examiner practiced under the guidance of an experienced examiner over a minimum of three 1-hour training sessions before testing study participants [35]. For knee extension, participants were seated with hips and knees in 90° and 60° flexion, respectively, and a handheld dynamometer placed 5 cm proximal to the distal tip of the lateral malleolus on the shin. For knee flexion, participants were in a prone position with the knee in 60° flexion and a dynamometer placed 5 cm proximal to the distal tip of the lateral malleolus on the calf. In all strength tests, the dynamometer was secured to the leg with an immovable strap. After a practice trial, participants completed three experimental trials consisting of 5 s of maximum effort pushing into the dynamometer followed by 15 s of rest. Peak isometric strength scores (N) were converted to peak torque (Nm; force x distance between joint line and dynamometer) and normalized to body weight (Nm/kg). Isometric knee muscle strength testing has sufficient intra- (pooled intraclass correlation coefficient > 0.90) [36,37] and inter-rater reliability (pooled intraclass correlation coefficient > 0.84) [36].

#### 2.5.5. Intermittent Knee Pain

Intermittent knee pain was assessed with the intermittent pain subscale of the ICOAP [38]. This subscale consists of six items that asks patients about “pain that comes and goes” over the past week. Each item is scored on a 5-point Likert scale, summed, and transformed to a subscale score ranging from 0–100 with lower scores indicating better outcomes. The ICOAP has not been evaluated in active youth populations but demonstrates sufficient internal consistency (Cronbach’s alpha 0.93) and test-retest reliability (intraclass correlation coefficient 0.85) in individuals with knee osteoarthritis [38].

#### 2.5.6. Physical Activity

Physical activity participation was self-reported using the GLTEQ [39]. Participants reported the number of 15-minute bouts of mild (minimal effort), moderate (not exhausting), and strenuous (heart beats rapidly) physical activity in which they engaged over a typical seven-day period. The total activity in metabolic equivalents (METs) is calculated by multiplying the number of mild, moderate, and strenuous bouts by 3, 5, and 9, respectively, and then summing these values. One MET equals the amount of energy expended by an individual seated at rest. The GLTEQ has been validated to assess physical activity [40,41]. Weekly moderate-to-strenuous METs were the focus of these analyses.

### 2.6. Data Analysis

Statistical analyses were performed using STATA (v12.1, Stata Corp., College Station, TX, USA). Descriptive statistics [median (range), proportion (95%CI)] were calculated for all participant characteristics and outcomes by study group (knee injury history or not).

Univariable linear regression models (95%CI), accounting for clustering on sex (female/male) and main sport type (e.g., soccer, ice hockey, basketball), were used to assess the association between previous injury history (yes/no) and HRQoL outcome (EQ-5D-5L index, EQ-VAS, and KOOS QOL). To better understand what factors might influence the relationship between injury history and HRQoL outcomes, separate multivariable linear regression models (95%CI), accounting for clustering on sex and main sport type, including sex, time since injury (years), type of injury (ACL tear or other), BMI (kg/m^2^), normalized peak knee extensor and flexor strength (Nm/kg), intermittent knee pain (ICOAP intermittent pain subscale), and moderate-to-strenuous physical activity (GLTEQ weekly METs) were considered. Biological sex [7,8], ACL tear [42,43], BMI [7], and knee strength [24] are established risk factors for osteoarthritis disease, whereas sex [16], age [17], BMI [21], pain [44], and physical activity [19], have been associated with generic and/or condition-specific HRQoL outcomes. Time since injury for uninjured participants was coded the same as that of matched injured participants on recruitment and indicate an equivalent injury-free time. Regression analyses began with models that included injury history (primary exposure variable), sex, time since injury, type of injury, BMI, knee extensor and flexor strength, intermittent knee pain, moderate-to-strenuous physical activity, and a two-way interaction term for injury history and sex. After evaluating the significance of the interaction term (i.e., likelihood ratio test, ≥0.05), we followed a backwards stepwise elimination approach where covariates with a *p*-value <0.05 were retained and the most parsimonious model was reported. All assumptions for linear regression analyses were assessed and met.

## 3. Results

A total of 253 participants were recruited, including 124 youth with a previous sport-related knee injury and 129 uninjured controls. The median age of the participants at follow-up was 23 years (range 14–29) and 55% of the participants were females (Table 1). Soccer was the most common pre-injury sport (35%) with ice hockey (21%), basketball (12%), skiing or snowboarding (8%), football (5%), rugby (4%), running (4%), volleyball (4%), dance or gymnastics (2%), horseback riding or rodeo (2%), baseball (1%), figure skating (1%), lacrosse (1%), and field hockey (1%) also identified. Of the injured group, 69 participants (56%) sustained a complete ACL tear, all of whom underwent ACL reconstruction. Twenty participants (16%) had meniscus injuries, 15 (12%) had other ligament injuries (i.e., grade I-II ACL or posterior cruciate ligament injury, grade I-III medial or lateral collateral ligament injury), 18 had a patellofemoral subluxation or dislocation (15%), and two (2%) had a fracture. The median time since injury was 6.7 years (range 2.9–11.6).

For generic HRQoL, the median EQ-5D-5L index score for the uninjured and injured participants was 0.911 (range 0.634–0.949) and 0.911 (range 0.561–0.949), respectively, and the median EQ-VAS score for the uninjured and injured participants was 85 (range 20–100) and 80 (range 10–100), respectively. For condition-specific HRQoL, uninjured participants had a median KOOS QOL score of 100 (range 83–100) whereas injured participants had a median score of 92 (range 64–100). One injured participant did not complete the EQ-5D-5L and three injured participants did not complete the EQ-VAS.

Univariable associations between injury history and HRQoL outcomes are summarized in Table 2. Injury history was not associated with EQ-5D-5L index (−0.0074, 95%CI −0.0238, 0.0089) or EQ-VAS scores (−3.82, 95%CI −8.77, 1.14). However, a negative association was found between injury history and KOOS QOL scores (−8.41, 95%CI −10.76, −6.06).

Multivariable linear regression models that considered the influence of sex, time since injury, type of injury, BMI, knee extensor and flexor strength, intermittent knee pain, and moderate-to-strenuous physical activity on the relationship between youth sport-related knee injury history and HRQoL outcomes are summarized in Table 3. Regardless of injury history, higher levels of intermittent pain (ICOAP) were associated with poorer generic HRQoL (EQ-5D-5L index −0.0024, 95%CI −0.0034, −0.0015) and, higher levels of moderate-to-strenuous physical activity (GLTEQ) were associated with better generic HRQoL (EQ-VAS 0.10, 95%CI 0.03, 0.17). A significant interaction between injury and sex suggested that injured males have slightly higher generic HRQoL (EQ-5D-5L index 0.0232, 95%CI 0.0042, 0.0422) than uninjured males. Finally, injury history (−5.14, 95%CI −6.90, −3.38), an ACL tear (−1.41, 95%CI −2.77, −0.06), and higher levels of intermittent pain (ICOAP −0.40, 95%CI −0.45, −0.36) were associated with lower condition-specific HRQoL (KOOS QOL). No other associations were found.

## 4. Discussion

Currently, most of what is known about risk factors for osteoarthritis is relative to markers of osteoarthritis disease (e.g., structural changes, biomechanics), not osteoarthritis illness (e.g., occupational or recreational time-loss, functional limitations, reduced HRQoL). This contrasts the fact that it is the illness that motivates people to seek healthcare and drives the individual and societal burden of osteoarthritis.

We present a novel examination of generic and condition-specific HRQoL in individuals with a previous youth sport-related knee injury compared to uninjured controls of similar age, sex, and sport. Our findings indicate that a 3–12 year history of a youth sport-related knee injury is not associated with generic HRQoL but is negatively associated with condition-specific HRQoL. Exploratory analyses revealed that more intermittent knee pain or less self-reported moderate-to-strenuous physical activity are associated with worse generic HRQoL, as measured by the EQ-5D-5L index and EQ-VAS scores, respectively. Regarding condition-specific HRQoL (KOOS QOL), injury history, a previous ACL tear, and more intermittent knee pain were associated with a poorer outcome. These data imply that there may be distinct determinants of generic and condition-specific HRQoL following a youth sport-related knee injury; therefore, these two outcomes, although related, need to be considered as unique parts of a broad construct. Furthermore, these preliminary analysis suggests that risk factors for osteoarthritis illness, including intermittent knee pain and participation in moderate-to-strenuous physical activity, may differ from osteoarthritis disease.

The finding that generic HRQoL was not associated with a 3–12 year history of a previous youth sport-related knee injury is consistent with a systematic review examining adults (*n* = 2493, mean age 34 years) at a mean of nine years following ACL reconstruction who reported similar or even better generic HRQoL (Short-Form 36) compared to population norms [14]. In contrast, individual analyses of Australian and Danish cohorts report lower generic HRQoL (Assessment of Quality of Life 4D instrument and EQ-5D-5L index, respectively) in relatively young individuals (*n* = 147, mean age 48 years) with radiographic osteoarthritis [47] and older individuals (*n* = 24,513, mean age 64.7 years) with radiographic and/or symptomatic osteoarthritis [48] compared to population norms [49]. With that said, few studies have assessed generic HRQoL in people with osteoarthritis while considering differences based on injury history.

It is plausible that the link between intermittent pain and generic HRQoL may be explained by its physical (e.g., sleep disturbance [50]), psychological (e.g., depression [51]), and social (e.g., activity or hobby limitation [50]) manifestations. Similarly, physical inactivity may influence generic HRQoL through its negative impact on physical (e.g., increased sedentary behaviour [52]), psychological (e.g., depression [53]), and social (e.g., isolation from sports/recreational community [54]) well-being. It is important to note that generic HRQoL could be influenced by factors not assessed here, including but not limited to other injuries, medical conditions (e.g., anxiety, depression, diabetes), and socioeconomic status. Although intermittent pain and moderate-to-strenuous physical activity may be determinants of generic HRQoL of young adults, the influence of many other aspects of physical, psychological, and social health should be investigated.

Current evidence indicates that individuals who have undergone an ACL reconstruction and those who have post-traumatic osteoarthritis may demonstrate deficits in condition-specific HRQoL. Specifically, reduced KOOS QOL scores have been observed at two years post-ACL reconstruction compared to uninjured controls (*n* = 120, mean age 19 years) [13] and persist up to 5–20 years post-op compared to population norms [14]. Reduced condition-specific HRQoL has also been identified by former collegiate athletes (*n* = 100, mean age 53.1 years) diagnosed with post-traumatic osteoarthritis compared to athletes with no knee surgery history [55]. Our findings expand beyond these observations to include individuals with a broad range of traumatic knee injuries that occurred in their youth. Taken together, these findings suggest that injury history is an important determinant of condition-specific HRQoL following a sport-related knee injury, regardless of age at injury and injury type.

Intermittent pain was identified as a possible determinant of condition-specific HRQoL, likely due to similar physical, psychological, and social manifestations as those mentioned above. It is also important to consider there is a moderate correlation between the KOOS QOL and pain subscales (Pearson’s correlation coefficient 0.603) [25], indicating that there may be some overlap of the constructs measured by these two subscales. Sustaining an ACL tear is a well-established risk factor for osteoarthritis disease [42] and, as our findings suggest, may also contribute to greater reductions in condition-specific HRQoL and possibly osteoarthritis illness. This is unsurprising as an ACL tear is associated with substantial physical impairments (e.g., knee muscle weakness [56]), psychological consequences (e.g., heightened fear of reinjury [57]), and social limitations (e.g., isolation from sport community [54]) as well as a relatively long rehabilitation period.

Aside from a past ACL tear, we found no association between other established risk factors for osteoarthritis disease (i.e., sex [7,8], BMI [7], knee strength [24]) and generic or condition-specific HRQoL, an important feature of osteoarthritis illness. Instead, what has emerged is the possible influence of intermittent pain and physical inactivity on generic and/or condition-specific HRQoL. These findings suggest that the forces leading to osteoarthritis disease and osteoarthritis illness following knee joint trauma may be somewhat distinct. They also suggest that intermittent pain and physical inactivity might be important targets for preventing osteoarthritis illness and highlight the need to consider both determinants of disease and illness when designing and evaluating prevention programs. Ultimately, a better understanding of the determinants of osteoarthritis disease and illness following a knee joint injury could contribute to the individualization of osteoarthritis prevention programs.

The strengths of this study are the inclusion of uninjured controls of similar age, sex, and sport exposure and a broad definition of knee injury (i.e., beyond an ACL tear) confirmed at the time of injury. In contrast, this study was not specifically powered for our research questions. However, these preliminary findings can be used to inform an adequately powered study to fully address related objectives. Many of the participants in the Alberta Youth PrE-OA cohort may be from middle-to-high socioeconomic status given the recruitment sources and their ability to access organized sport, post-secondary education, and healthcare which limits the generalizability of our findings. Future studies should seek diverse and inclusive samples to better understand what happens to people from all backgrounds following a youth sport-related knee injury. It is important to highlight that the KOOS QOL subscale only consists of four items which may not capture the breadth and complexity of condition-specific HRQoL. Although only limited to people with an ACL injury, a possible alternative is the Anterior Cruciate Ligament-Quality of Life questionnaire (ACL-QOL) [58] which comprehensively assesses multiple domains of condition-specific HRQoL (i.e., symptoms and physical complaints, work-related concerns, recreational activities and sports participation, lifestyle, and social and emotional). Using a self-report measure of physical activity also introduces possible recall bias. When possible, accelerometry should be utilized as it is a more valid measure of physical activity. Lastly, only data on biological sex is available for the Alberta Youth PrE-OA study. Arguably, one’s biological sex and socially constructed gender could influence HRQoL and, therefore, both the influence of sex and gender should be examined going forward [59].

More research is required to better understand osteoarthritis illness, particularly in individuals who have sustained a previous youth sport-related knee injury. Further investigation of both generic and condition-specific HRQoL outcomes as it relates to osteoarthritis prevention is needed as they likely represent related yet separate constructs. Future studies should confirm and continue to explore the determinants of generic and condition-specific HRQoL and assess how these outcomes change in the short-, medium-, and long-term after a youth sport-related knee injury. In addition to building upon previous evidence, we recommend engaging patients as research partners to ensure relevant constructs related to HRQoL and osteoarthritis illness are examined. This information can be leveraged to develop osteoarthritis prevention programs that target aspects of both osteoarthritis disease and illness.

## 5. Conclusions

The findings of this study suggest that generic and condition-specific HRQoL are distinct from one another and, therefore, both should be measured in research and clinical practice. Injury history appears to be associated with condition-specific but not generic HRQoL. Based on exploratory analyses, intermittent knee pain and moderate-to-strenuous physical activity may be factors that influence generic HRQoL whereas injury history, injury type, and intermittent knee pain may be factors that influence condition-specific HRQoL. Targeting intermittent knee pain and physical activity, particularly in youth and young adults who have sustained an ACL injury, may help optimize HRQoL. This study provides preliminary evidence that risk factors for osteoarthritis illness may differ from osteoarthritis disease. This finding can inform the design of future studies and osteoarthritis prevention strategies.

## Figures and Tables

**Table 1 ijerph-18-06877-t001:** Participant characteristics, outcomes, and covariates by study group.

Characteristic	Uninjured (*n* = 129)	Injured (*n* = 124)
Sex (% female, 95%CI)	56 (47, 64)	53 (44, 62)
Age at injury (years)	-	16 (9–19)
Age at follow-up (years)	23 (14–29)	22 (16–29)
Time since injury (years)	-	6.7 (2.9–11.6)
Type of injury (% ACL tear, 95%CI)	-	56 (47, 64)
Subsequent injury (% yes, 95%CI)	1 (0, 7)	27 (29, 46)
Subsequent surgery (% yes, 95%CI)	-	21 (15, 29)
Radiographic osteoarthritis (% yes, 95%CI) *	0 (0, 0)	7 (3, 15)
MRI-defined osteoarthritis (% yes, 95%CI) †	3 (1, 10)	28 (19, 38)
Main sport (% soccer, 95%CI)	35 (27, 44)	35 (27, 44)
Sport participation in last 12 months (% yes, 95%CI)	95 (90, 98)	89 (81, 93)
EQ-5D-5L index	0.911 (0.634–0.949)	0.911 (0.561–0.949)
EQ-VAS	85 (20–100)	80 (10–100)
KOOS QOL	100 (83–100)	92 (64–100)
BMI (kg/m^2^)	23.5 (18.1–33.1)	24.8 (18.6–38.9)
Knee extensor strength (Nm/kg)	1.92 (0.73–4.21)	1.84 (0.40–3.53)
Knee flexor strength (Nm/kg)	1.09 (0.38–2.08)	0.95 (0.37–2.09)
ICOAP intermittent pain	0 (0–33)	0 (0–54)
GLTEQ moderate-to-strenuous physical activity (METs/week)	45 (0–93)	42 (4–136)

Values represent median (range) unless otherwise indicated. Subsequent injury = any tibiofemoral or patellofemoral injury that resulted in seeking medical attention and time-loss from sport participation. Subsequent surgery = any surgery to the index or non-index knee during the follow-up period. Radiographic osteoarthritis of index knee = grade ≥ 2 on the Kellgren-Lawrence Grading System [45]. MRI-defined osteoarthritis of index knee = met criteria for tibiofemoral (medial or lateral compartment), mixed tibiofemoral, or patellofemoral MRI-defined osteoarthritis as per Hunter et al. (2011) [46]. * Data available for 86 uninjured and 84 injured participants. † Data available for 88 uninjured and 87 injured participants. ACL, anterior cruciate ligament; BMI, body mass index; EQ-5D-5L, EuroQoL five-dimension, five-level; EQ-VAS, EuroQoL visual analog scale; GLTEQ, Godin Leisure-Time Exercise Questionnaire; ICOAP, Intermittent and Constant Osteoarthritis Pain Score; kg, kilogram; KOOS QOL, Knee injury and Osteoarthritis Outcome Score quality of life subscale; m, metre; MET, metabolic equivalent; MRI, magnetic resonance imaging; *n*, number of participants; Nm, Newton-metre; 95%CI, 95% confidence interval.

**Table 2 ijerph-18-06877-t002:** Univariable linear regression models for injury history and HRQoL outcomes.

Model	*n*	Injury History *	*r* ^2^
1. EQ-5D-5L	252	−0.0074 (−0.0238, 0.0089)	0.005
2. EQ-VAS	250	−3.82 (−8.77, 1.14)	0.022
3. KOOS QOL	253	**−8.41 (−10.76, −6.06)**	0.305

Values represent coefficient and 95%CI. All models accounted for clustering by sex and sport. Bolded font represents 95%CI does not encompass zero. * Reference = uninjured participants. EQ-5D-5L, EuroQoL five-dimension, five-level; EQ-VAS, EuroQoL visual analog scale; KOOS QOL, Knee injury and Osteoarthritis Outcome Score quality of life subscale; *n*, number of participants; *r*^2^, coefficient of determination; 95%CI, 95% confidence interval.

**Table 3 ijerph-18-06877-t003:** Multivariable linear regression models for injury history and HRQoL outcomes considering determinants of HRQoL and osteoarthritis disease.

Model	Injury History *	Sex †	Time Since Injury	ACL ‡	BMI (kg/m^2^)	Extensor Strength (Nm/kg)	Flexor Strength (Nm/kg)	ICOAP	GLTEQ (MET/wk)	Injury × Sex	*r* ^2^
1. EQ-5D-5L	−0.0032 (−0.0170, 0.0107)	−0.0090 (−0.0227, 0.0047)						**−0.0024 (−0.0034, −0.0015)**		**0.0232 (0.0042, 0.0422)**	0.220
2. EQ-VAS	−3.47 (−7.98, 1.04)								**0.10 (0.03, 0.17)**		0.047
3. KOOS QOL	**−5.14 (−6.90, −3.38)**			**−1.41 (−2.77, −0.06)**				**−0.40 (−0.45, −0.36)**			0.587

Values represent coefficient and 95%CI. All models accounted for clustering by sex and sport. Bolded font represents 95%CI does not encompass zero. Shaded cells represent variables that were removed due to lack of evidence of modification or confounding. * Reference = uninjured participants. † Reference = female sex. ‡ Reference = no ACL tear. ACL, anterior cruciate ligament tear; BMI, body mass index; EQ-5D-5L, EuroQoL five-dimension, five-level; EQ-VAS, EuroQoL visual analog scale; GLTEQ, Godin Leisure-Time Exercise Questionnaire moderate-to-strenuous physical activity; ICOAP, Intermittent and Constant Osteoarthritis Pain intermittent pain subscale; kg, kilogram; KOOS QOL, Knee injury and Osteoarthritis Outcome Score quality of life subscale; m, metre; MET, metabolic equivalent; Nm, Newton-metre; *r*^2^, coefficient of determination; wk, week; 95% CI, 95% confidence interval.

## Data Availability

The data presented in this study are available from the corresponding author on reasonable request.

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
