# Peer review of "What Does the Future Hold? Health-Related Quality of Life 3–12 Years Following a Youth Sport-Related Knee Injury"

_ijerph, 2021, doi:10.3390/ijerph18136877_

Round 1
Reviewer 1 Report
Thank you for the opportunity to review this manuscripts. The authors should be commended on conducting exciting research related to youth and osteoarthritis. The PRE-OA cohort study has potential to bring much insight into the impact of youth injury on the lives of people in the longer term.
In general, the manuscript is very well written and easy to follow. There are numerous details about the population, methods, instruments, and analyses. One typo was found on line 128 (remove "is" before "on"). The sentence starting on 335 doesn't read clearly - suggest revisiting to improve clarity of point.
The two primary comments and areas to adjust relate to Table 3 and to the idea of osteoarthritis illness vs. osteoarthritis disease.
Table 3: In general, the table is thoughtfully constructed to show the results of the multivariable analysis. However, in some ways as a reader and someone with less knowledge of linear regression models on how to use that information. The values, particularly, for the EQ-5D-5L are very small (and this is understood given the range of scale score on this measure) and it's intuitively hard to see what value or meaning that has in relation to someone's quality of life. Is there a way to better present these data or to describe them in the text and address the small coefficients? To use and understand these findings clinically, more assistance to the reader in terms of what this means may be helpful.
The other question is related to the idea of osteoarthritis illness vs. disease. The idea that there are two concepts related to osteoarthritis is brought up in the introduction very briefly. The discussion talks much more or refers to "osteoarthritis illness" much more. Is there a way to elaborate or expand on this idea more in the introduction to better set up the discussion? The discussion is very well written and the ideas are interesting, especially the discussion related to osteoarthritis illness. However, the concept of osteoarthritis illness in contrast to osteoarthritis disease needs a bit more building or explanation - and more citations. On a brief search, not much was found related to the idea of osteoarthritis illness. It makes sense that there are factors related to illness (HRQOL) and to disease and that these can be different. More discussion on these two concepts would strengthen the manuscript. Additionally, because of the emphasis on osteoarthritis illness in the discussion, it begs the question as to whether this should be a focus of the purpose of the study? The objective of the study is to assess generic and specific HRQOL and there is no mention of osteoarthritis illness as part of the purpose. If it's not part of the purpose, that's fine, and I would suggest working to ensure the purpose aligns with the tone of the discussion.
Again, the manuscript was well written and detailed with good integration of literature in the discussion to build some of the points.
Author Response
Thank you for your feedback on this manuscript. Please find our responses to your comments attached.

Reviewer 2 Report
This is an interesting study on HRQoL in individuals with a 3-12 year history of a youth sport-related knee injury and respective influence factors. The study is well-conducted and presented.
I have no concerns. It can be accepted in the present form.
Author Response

(The authors gave the same response as above.)

Reviewer 3 Report
Proposed adaptations according to annexed notes

Author Response

(The authors gave the same response as above.)
